# Did COVID-19 Pandemic Change People’s Physical Activity Distribution, Eating, and Alcohol Consumption Habits as well as Body Mass Index?

**DOI:** 10.3390/ijerph182312405

**Published:** 2021-11-25

**Authors:** Albertas Skurvydas, Ausra Lisinskiene, Marc Lochbaum, Daiva Majauskiene, Dovile Valanciene, Ruta Dadeliene, Natalja Fatkulina, Asta Sarkauskiene

**Affiliations:** 1Education Academy, Vytautas Magnus University, K. Donelaičio Str. 58, 44248 Kaunas, Lithuania; albertas.skurvydas@vdu.lt (A.S.); marc.lochbaum@ttu.edu (M.L.); daiva.majauskiene@vdu.lt (D.M.); dovile.valanciene@gmail.com (D.V.); 2Department of Rehabilitation, Physical and Sports Medicine, Institute of Health Sciences, Faculty of Medicine, Vilnius University, 21/27 M.K. Čiurlionio St., 03101 Vilnius, Lithuania; ruta.dadeliene@mf.vu.lt; 3Institute of Educational Research, Education Academy, Vytautas Magnus University, K. Donelaičio Str. 58, 44248 Kaunas, Lithuania; 4Department of Kinesiology and Sport Management, Texas Tech University, Lubbock, TX 79409, USA; 5Department of Physical and Social Education, Lithuanian Sports University, Sporto Str. 6, 44221 Kaunas, Lithuania; 6Institute of Sport Science and Innovations, Lithuanian Sports University, Sporto Str. 6, 44221 Kaunas, Lithuania; 7Institute of Health Sciences, Faculty of Medicine, Vilnius University, 21/27 M.K. Čiurlionio St., 03101 Vilnius, Lithuania; natalja.fatkulina@mf.vu.lt; 8Department of Sports, Recreation and Tourism, Klaipėda University, Herkaus Manto St. 84, 92294 Klaipėda, Lithuania; asta.sarkauskiene@ku.lt

**Keywords:** COVID-19, physical activity, eating, alcohol consumption habits, body mass index

## Abstract

This cross-sectional study aimed to evaluate whether COVID-19 had an impact on people’s (aged 18–74) physical activity distribution, eating, and alcohol consumption habits as well as body mass index. We interviewed 6369 people (4545 women and 1824 men) in Lithuania before the COVID-19 pandemic started and 2392 during COVID-19 (1856 women and 536 men). They were aged 18–74 years. We found that both genders had not stopped their physical activity (PA) completely because of lockdown imitations (for example, prohibition from attending sport clubs), but they started doing different physical exercises at sport clubs. We determined the PA distribution according to the Danish Physical Activity Questionnaire (DPAQ). Despite increases in independent PA and the quantity of light PA, the amount of total energy used in metabolic equivalent of task (MET) units per day decreased significantly for both genders irrespective of age. Although the amounts of sedentary behavior, moderate PA (MPA), vigorous PA (VPA) or a combination of MPA and VPA (MVPA) did not change significantly. Surprisingly, lockdown reduced the duration of sleep for older women but increased their amount of intense VPA (>6 METs). However, the amount of intense VPA decreased for men. Both genders reported overeating less during the pandemic than before it, but did not start consuming more alcohol, and their body mass index did not change. Thus, the COVID-19 in Lithuania represented ‘good stress’ that mobilized these individuals to exercise more independently and overeat less.

## 1. Introduction

There is growing evidence that various forms and doses of physical activity (PA) are effective in combating many chronic diseases [1,2]. Thus, the effect of PA on various body functions is quite specific in that it depends in a nonlinear manner on muscle work intensity, duration, and load ‘doses’ [2,3]. In addition, the health benefits of PA also depend on the subject’s age, gender, health status, and body mass index (BMI) [2,4,5,6]. It is clear that insufficient PA results in increased obesity and later systematic inflammation, causing many chronic diseases [1,7]. Guthold et al. (2018) [8] summarized the dynamics of 1.9 million reports of human physical inactivity from 2001 to 2016 in developed European countries. Clearly, this lack of PA increased significantly for both men and women. Obesity and low PA can be interrelated: thus, low PA stimulates obesity, and this, in turn, reduces an individual’s motives to be physically active [9] and stimulates more frequent overeating because of an inability to control the appetite [10].

A recent meta-analysis showed that the COVID-19 pandemic and associated quarantines and lockdowns in Lithuania (abbreviated hereafter as ‘COVID-19’) reduced opportunities to participate in sport or exercise for leisure and health purposes. However, it also provided many individuals with increased time available to engage in more regular PA and exercise [11]. The closure of gyms, indoor athletic and leisure centers, as well as the cancellation of recreational sport, and limitations on all but essential travel have likely caused a decline in the amount of PA [12]. Interviews with 565 sportspeople with high levels of mastery revealed that the isolation associated with COVID-19 had worsened the quality of their training, reduced their daily PA, prolonged the duration of sleep, and worsened their mental health [13].

During COVID-19 episodes, snacking and alcohol consumption increase [14]. Thus, the prevalence of physical inactivity and sedentary behavior (SB) increased in all population subgroups during the COVID-19 pandemic in Brazil [15]. An increase in body weight was shown in about half of the respondents during the first COVID-19 outbreak in Poland. This study group showed a decrease in PA and increases in overall food and high-energy product consumption [16].

Despite the above-mentioned studies, it is not yet clear how COVID-19 has changed the structure of PA, for example, the distribution of SB, light PA (PA), moderate PA (MPA), or vigorous PA (VPA). Moreover, it is not clear whether the changes in types of PA, or in sleeping, eating, and alcohol consumption behaviors depend on gender and age during lockdowns. In this sense, it was important to find out how COVID-19 Pandemic change people’s physical activity distribution, eating, and alcohol consumption habits as well as body mass index? To our knowledge, this is the first study in Lithuania that determined physical activity doses, eating and alcohol consumption habits and body mass index in various age groups of adults before and during COVID-19 and revealed an overall picture of the whole of Lithuania in the context of the research problem. Therefore, this cross-sectional study aimed to evaluate whether COVID-19 had an impact on people’s (aged 18–74) Physical Activity Distribution, Eating, and Alcohol Consumption Habits as Well as Body Mass Index.

## 2. Materials and Methods

### 2.1. Participants

Participants were recruited before COVID-19 in Lithuania. This baseline group included 6369 people (4545 women and 1824 men). During COVID-19, we recruited 2392 additional subjects (1856 women and 536 men). Participants were aged 18–74 years. There were 78.3% and 79.5% people with higher and university-level education in the groups recruited before and during COVID-19, respectively. Of these, 83.4% and 83.1% were city-dwellers, respectively. The mean ages before and during COVID-19 were 37.9 ± 11.8 and 38.4 ± 12.6 years, respectively.

### 2.2. Procedure

The first research was performed from October 2019 to June 2020. The second research was performed from November 2020 to March 2021. The subjects were included to represent the Lithuanian population, and participation was anonymous, so data collection and handling were confidential. We used an online survey application to collect information (https://docs.google.com/forms/ (accessed on 6 March 2021)). All participants completed the online questionnaires, and the online survey link was also distributed through social media (Facebook) and personal contacts (WhatsApp). The web-based open E-survey research was submitted for the approval by the Ethic committee.

Besides, we ensured that the study was performed according to the principles laid by, declaration of Helsinki (Revised 2013) and National guidelines for biomedical and health research involving human participants (2017). The purpose of the survey, introduction and about the length of the survey was added within the web-based open E-survey. A successful return of completed survey was considered as consent by the participant.

### 2.3. Measures

We applied a quantitative, cross-sectional study design. The following instruments have been used in conducting this study: The DPAQ has been adapted from the International Physical Activity Questionnaire (IPAQ; https://loinc.org/77582-5/ (accessed on 25 August 2019)) and differs from it by referring to the subject’s PA of the last 24 h for 7 consecutive days, instead of simply the last 7 days. The chosen activities are listed in the PA scale at nine levels of physical exertion, ranging from sleep or SB (0.9 MET) to strenuous activities (>6 METs). Each level in terms of metabolic activity of task (MET) values (A = 0.9, B = 1.0, C = 1.5, D = 2.0, E = 3.0, F = 4.0, G = 5.0, H = 6.0, and I > 6) is described in the DPAQ by examples of specific activities for that particular level and by a small drawing. The PA scale was constructed so that the number of minutes (15, 30, or 45) and hours (1–10) spent at each MET activity level on an average 24-h weekday could be reported. This allowed for a calculation of the total MET time, representing 24 h of sleep, work, and leisure time on an average weekday [17,18].

We calculated how much energy in terms of METs was consumed per day during sleeping, SB (0.9–1.5 METs), light intensity PA (LPA; >1.5 <3 METs), moderate intensity PA (MPA; 3 to <6 METs), vigorous intensity PA (VPA; >6 METs). We also combined MPA and VPA as MVPA, and we calculated how many METs were wasted when the intensity was >6 METs: as extra vigorous PA (VPAextra).

### 2.4. Data Analysis

Data are reported as the mean ± standard error. The data were tested for normality using the Kolmogorov–Smirnov test, and all data were found to be normally distributed. We also performed calculations to evaluate the observed power (OP) of findings, partial eta squared values $\left(\eta_{P}^{2}\right)$ and chi-square (χ^2^) values. Univariate two-way analysis of variance (ANOVA) was performed to determine whether there was any interaction between the two independent variables and the dependent variable. If significant effects were found, Tukey’s post hoc adjustment was used for multiple comparisons within each repeated measures ANOVA. For all tests, statistical significance was defined as *p* < 0.05. Statistical analyses were performed using IBM SPSS Statistics software (version 22; IBM Corp., Armonk, NY, USA).

## 3. Results

### 3.1. In What Way Did COVID-19 Change the Forms of PA for Men and Women?

The numbers of nonexercising people (both genders) did not change during COVID-19, but there was redistribution: thus, the numbers of people exercising at sport centers reduced significantly (*p* < 0.001), but the numbers exercising independently increased (female, *p* < 0.001, chi-square value 400.1; male, *p* < 0.001, chi-square value 67.5; Table 1).

### 3.2. Effect of COVID-19 on the PA Structure for Men and Women

Total METs decreased significantly for men and women because of COVID-19 (COVID-19 effect, *p* = 0.001, $\eta_{P}^{2}$ = 0.001, OP = 0.93; age effect, *p* < 0.0001, $\eta_{P}^{2}\;$= 0.004, OP = 1; gender effect, *p* < 0.0001, $\eta_{P}^{2}$ = 0.11, OP = 1; interaction effect, nonsignificant, n.s.; Figure 1). Moreover, MVPA did not decrease significantly because of COVID-19 (COVID-19 effect, *p* = 0.086, $\eta_{P\;}^{2}$ = 0.001, OP = 0.61; age effect, *p* < 0.0001, $\eta_{P}^{2}$ = 0.005, OP = 1; gender effect, *p* < 0.0001, $\eta_{P}^{2}$ = 0.02, OP = 1; interaction effect, n.s.). Thus, COVID-19 reduced total METs but did not change MVPA irrespective of age, although the MVPA of women aged 18–25 y decreased significantly (*p* = 0.011).

LPA decreased significantly because of COVID-19 (COVID-19 effect, *p* < 0.0001, $\eta_{P}^{2}\;$= 0.002, OP = 1; age effect, *p* < 0.0001, $\eta_{P}^{2}=$ 0.006, OP = 1; gender effect, *p* < 0.0001, $\eta_{P}^{2}=$ 0.017, OP = 1; interaction effect, n.s.; Figure 2). Interestingly, VPAextra increased for women (26–40 and ≥41 years) and decreased for men because of COVID-19 (interaction effect, COVID-19’ gender: *p* < 0.0001, $\eta_{P}^{2}\;$= 0.009, OP = 1). Neither VPA (COVID-19 effect, *p* = 0.076, $\eta_{P}^{2}=$ 0.001, OP = 0.51; age effect, *p* < 0.0001, $\eta_{P}^{2}\;$= 0.007, OP = 1; gender effect, *p* < 0.0001, $\eta_{P\;}^{2}\;$= 0.034, OP = 1; interaction effect, n.s.) nor MPA changed significantly because of COVID-19 (COVID-19 effect, *p* = 0.27, $\eta_{P}^{2}$ < 0.0001, OP = 0.21; age effect, *p* < 0.0001, $\eta_{P}^{2}$ = 0.009, OP = 1; gender effect, *p* = 0.21; $\eta_{P}^{2}$ < 0.0001, OP = 0.27, interaction effect, n.s.; Figure 2).

### 3.3. Did COVID-19 Affect the Duration of Sleep and SB?

The duration of sleep (number of METs used) among women aged 26–40 and ≥41 years decreased significantly because of COVID-19 (interaction effect of COVID-19’ age’ gender: *p* = 0.001, $\eta_{P}^{2}=$ 0.002, OP = 0.930), but it did not change the SB METs (COVID-19 effect, *p* = 0.77, $\eta_{P\;}^{2}$ < 0.0001, OP = 0.11; age effect, *p* = 0.24, $\eta_{P}^{2}\;$< 0.0001, OP = 0.29; gender effect, *p* = 0.39, $\eta_{P\;}^{2}$ < 0.0001, OP = 0.21; interaction effect, n.s.; Figure 3).

### 3.4. Effect of COVID-19 on Healthy Eating and Alcohol Drinking

Alcohol consumption did not change significantly during COVID-19 for women and men (chi-square and *p* values were 12.1 and 0.063 for women, versus 4.8 and 0.56 for men, respectively; Table 2). Both the men and the women reported overeating less during COVID-19 (chi-square and *p* values for women and men were 8.0 and 0.018; 7.4 and 0.018, respectively; Table 3).

### 3.5. Effect of COVID-19 on BMI

Surprisingly, the BMI values of both genders did not change significantly during COVID-19 (COVID-19 effect, *p* = 0.8, $\eta_{P}^{2}\;$< 0.0001, OP = 0.057; age effect, *p* < 0.0001, $\eta_{P}^{2}$ = 0.036, OP = 1; gender effect, *p* < 0.0001, $\eta_{P}^{2}$ = 0.023, OP = 1; interaction effect, n.s.; Figure 4). The effect of COVID-19 on the BMI distribution was not significant (chi-square and *p* values for women and men were 0.83 and 0.93, and 7.5 and 0.11, respectively; Table 2). However, the percentage of men with a BMI 25–29.9 kg/m^2^ increased significantly during COVID-19 (*p* < 0.05) and that of men with a BMI 18–24.9 kg/m^2^ decreased (*p* < 0.05; Table 4).

## 4. Discussion

Clearly, our findings show that women and men had not stopped PA completely because of lockdown limitations during the COVID-19 (for example, prohibition from attending sport clubs), but they started doing different physical exercises independently, instead of undertaking PA at sport clubs. Despite the increase in independent PA during COVID-19 and the quantity of light PA, the amount of total energy used per day decreased significantly for both genders irrespective of age, although the distributions of SB, MPA, VPA, and MVPA did not change significantly. COVID-19 reduced the duration of sleep for elderly women and increased their VPAextra METs. However, VPAextra decreased for men because of COVID-19. Both genders reported overeating less during COVID-19 than before it, but neither started consuming more alcohol. Thus, to our knowledge, our research showed for the first time how energy usage changed because of the COVID-19 (sleeping, SB, LPA, MPA, VPA, and VPAextra) depending on age and gender. Decreased overeating certainly affected the BMI distribution among men, although the overall BMI did not change because of COVID-19. The strength of our research is that we had one of the biggest samples reported when analyzing PA before and during COVID-19. Moreover, we think we chose the main aspects of lifestyle medicine (e.g., healthy eating, PA type, sleeping, and tobacco/alcohol consumption) ensuring a healthy lifestyle before and during COVID-19 [19]. It is known that changes in diet, sleeping quality, and types of PA are associated with differences in negative mood during COVID-19 lockdowns [20].

Our research showed clearly that the COVID-19 in Lithuania involved ‘good stress’ that mobilized people to exercise more independently and overeat less. Similar conclusions were Zhang et al. (2021) [21]. also determined during the COVID-19 pandemic in the USA, where healthy eating behavior and PA increased. However, there were also increases in addictive lifestyle behaviors including the abuse of alcohol, tobacco, and vaping [20]. ‘Good stress’ stimulated men and women to exercise independently more and not to reduce the most important form of MVPA, which is the most important guarantee of health improvement [2,22]. As the reported use of MVPA did not change during COVID-19 in our study, it indicated good health among our subjects. This is because others have shown that lockdown and quarantining have negative effects on people’s psychological health [23], but PA reduces the symptoms of depression and anxiety during COVID-19 [24]. In addition, our research did not show any increases in BMI during COVID-19, which should help in reducing the risk of severe COVID-19 illness [25].

Our findings do not agree with those of other studies, that people started drinking more alcohol because of COVID-19 isolation [12], that their BMI increased [16], and that episodes of SB became longer [15]. Epidemiological studies have indicated that depression and obesity have a strong bidirectional relationship; thus, increases in BMI increase the risk for developing depression and vice versa, so that individuals with depression tend to have a high BMI [26].

## 5. Limitations and Directions for Future Research

The main limitation of this study was the self-administered DPAQ questionnaire because it might have overestimated the various types of PA slightly. Thus, Danish research has shown that the DPAQ overestimates the time spent on light, moderate, and vigorous intensity PA and underestimates the time spent on SB [27]. In addition, as others have observed, it is difficult to compare PA data between study groups because of the variety of methodologies used [2,22,28]. Another limitation is that we were unable to study the same subjects before and during COVID-19 in our country. Moreover, we interviewed 6369 people (4545 women and 1824 men) in Lithuania before the COVID-19 pandemic and 2392 during COVID-19. The study limitation is that participants were not the same sample before and during the COVID-19. Therefore, the sample of the study before and during the COVID-19 was appropriate and led to evaluate adequately people PA, Eating, Alcohol Consumption Habits as Well as Body Mass Index tendencies in Lithuania.

## 6. Conclusions

Both women and men in this survey in Lithuania had replaced their form of PA at sport clubs with different physical exercises because of the limitations of the COVID-19. Despite of age, the quantity of energy used per day and the quantity of light PA decreased for both genders, although the METs used in SB, MPA, VPA, and MVPA did not change significantly. COVID-19 reduced the duration of sleep for elderly women (the duration of sleep of other subjects did not change) and increased the amount of VPAextra, whereas this measure decreased for men.

We conclude that during COVID-19 in our country represented ‘good stress’ that mobilized people to exercise more, independently or in sport clubs, overeat less, did not start consuming more alcohol.

## Figures and Tables

**Figure 1 ijerph-18-12405-f001:**
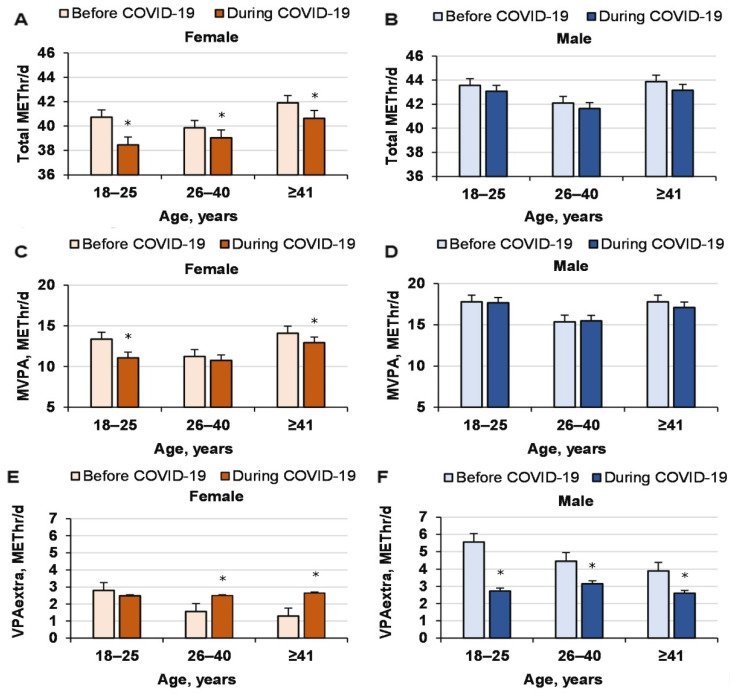
Total energy used per day in METs and during MVPA before and during COVID-19 for men and women of different ages. * compared to before COVID-19.

**Figure 2 ijerph-18-12405-f002:**
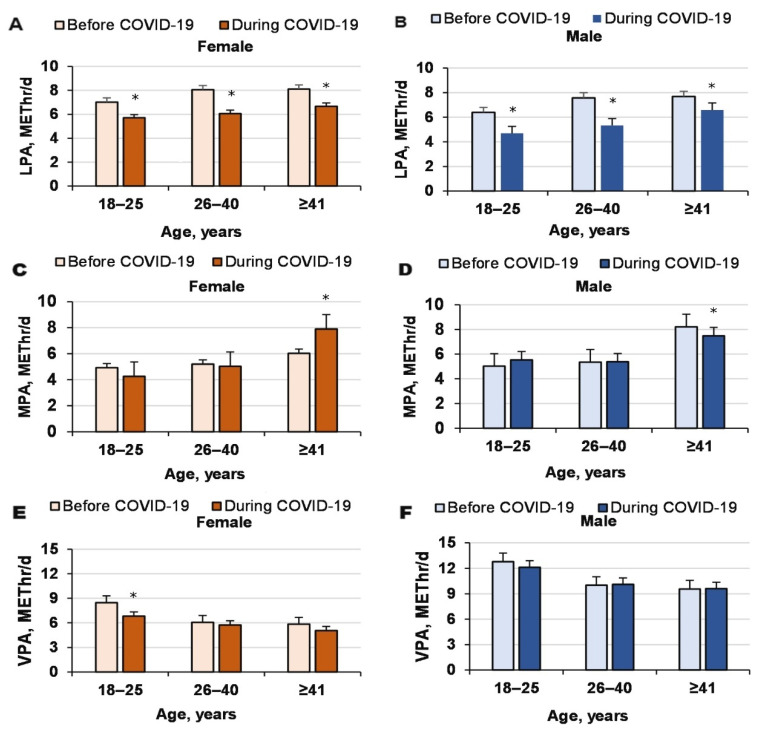
PA distributions before and during COVID-19 for men and women of different ages. * compared to before COVID-19.

**Figure 3 ijerph-18-12405-f003:**
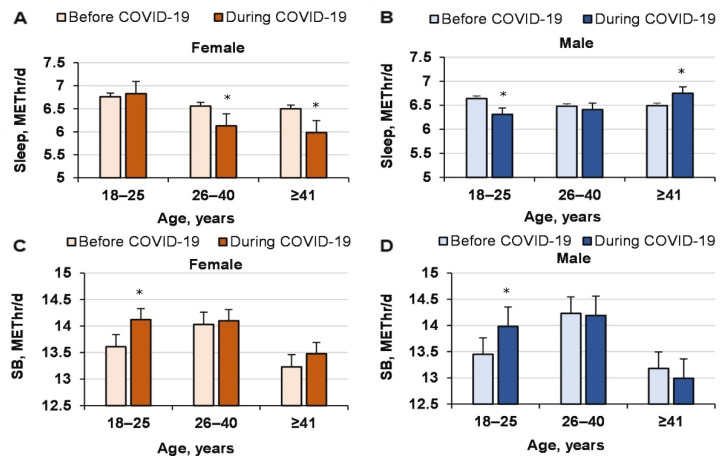
Sleep and sedentary behavior (SB) METs before and during COVID-19 for men and women of different ages. * compared to before COVID-19.

**Figure 4 ijerph-18-12405-f004:**
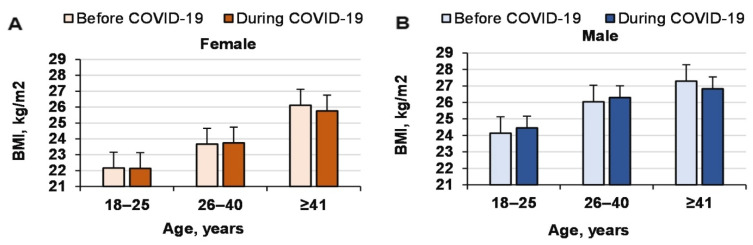
Changes in BMI before and during COVID-19 for men and women of different ages.

**Table 1 ijerph-18-12405-t001:** Choice of the forms of PA for men and women before and during COVID-19.

Gender	Variables	COVID-19			
		Before		During	
		Count	%	Count	%
Female	I don’t exercise	1726 _a_	38.0%	727 _a_	39.2%
	I’m in a professional sport	154 _a_	3.4%	49 _a_	2.6%
	I exercise by myself	1328 _a_	29.2%	919 _b_	49.5%
	I exercise in a gym/health center	1337 _a_	29.4%	161 _b_	8.7%
Male	I don’t exercise	394 _a_	21.6%	138 _b_	25.7%
	I’m in a professional sport	139 _a_	7.6%	50 _a_	9.3%
	I exercise by myself	875 _a_	48.0%	311 _b_	58.0%
	I exercise in a gym/health center	416 _a_	22.8%	37 _b_	6.9%

Each subscript letter denotes a subset of COVID-19 MET categories that do not differ significantly from each other at *p* < 0.05.

**Table 2 ijerph-18-12405-t002:** Alcohol consumption before and during COVID-19.

Gender	Variables	COVID-19			
		Before		During	
		Count	%	Count	%
Female	I don’t drink alcohol at all	669 _a_	14.7%	303 _a_	16.3%
	I drink alcohol several times a year	1464 _a_	32.2%	561 _a_	30.2%
	I drink alcohol once a month	759 _a_	16.7%	289 _a_	15.6%
	I drink alcohol several time a month	933 _a_	20.5%	347 _a_	18.7%
	I drink alcohol once a week	403 _a_	8.9%	201 _b_	10.8%
	I drink alcohol few times a week	271 _a_	6.0%	135 _a_	7.3%
	I drink alcohol every day	46 _a_	1.0%	20 _a_	1.1%
Male	I don’t drink alcohol at all	255 _a_	14.0%	94 _b_	17.6%
	I drink alcohol several times a year	444 _a_	24.3%	124 _a_	23.1%
	I drink alcohol once a month	282 _a_	15.5%	84 _a_	15.7%
	I drink alcohol several time a month	389 _a_	21.3%	106 _a_	19.8%
	I drink alcohol once a week	217 _a_	11.9%	65 _a_	12.1%
	I drink alcohol few times a week	189 _a_	10.4%	51 _a_	9.5%
	Every day	48 _a_	2.6%	12 _a_	2.2%

Each subscript letter denotes a subset of COVID-19 categories whose column proportions do not differ significantly from each other at the *p* < 0.05 level.

**Table 3 ijerph-18-12405-t003:** Overeating behavior before and during COVID-19.

Gender	Variables	COVID-19			
		Before		During	
		Count	%	Count	%
Female	I overeat seldom	2837 _a_	62.4%	1155 _a_	62.2%
	I overeat often	883 _a_	19.4%	318 _b_	17.1%
	I never overeat	825 _a_	18.2%	383 _b_	20.6%
Male	I overeat seldom	1190 _a_	65.2%	357 _a_	66.6%
	I overeat often	302 _a_	16.6%	65 _b_	12.1%
	I never overeat	332 _a_	18.2%	114 _a_	21.3%

Each subscript letter denotes a subset of COVID-19 categories whose column proportions do not differ significantly from each other at the *p* < 0.05 level.

**Table 4 ijerph-18-12405-t004:** Changes in BMI distribution during COVID-19.

Gender	BMI	COVID			
		Before		During	
		Count	%	Count	%
Female	<18.00	111 _a_	2.4%	49 _a_	2.6%
	18–24.9	2929 _a_	64.5%	1198 _a_	64.5%
	25–29.9	1032 _a_	22.7%	424 _a_	22.8%
	30–34.9	347 _a_	7.6%	140 _a_	7.5%
	35 and more	126 _a_	2.8%	45 _a_	2.4%
Male	<18.00	6 _a_	0.3%	2 _a_	0.37%
	18–24.9	816 _a_	44.7%	209 _b_	38.99%
	25–29.9	803 _a_	44.0%	263 _b_	49.07%
	30–34.9	160 _a_	8.8%	54 _a_	10.08%
	35 and more	39 _a_	2.1%	8 _a_	1.5%

Each subscript letter denotes a subset of COVID-19 categories whose column proportions do not differ significantly from each other *p* < 0.05 level.

## Data Availability

The data is available upon request.

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
