# Peer review of "Did COVID-19 Pandemic Change People’s Physical Activity Distribution, Eating, and Alcohol Consumption Habits as well as Body Mass Index?"

_ijerph, 2021, doi:10.3390/ijerph182312405_

Round 1
Reviewer 1 Report
Dear authors and editor,
The manuscript titled "Did the First Wave of the COVID-19 Pandemic Change People’s Physical Activity Distribution, Eating, and Alcohol
Consumption Habits as Well as Body Mass Index?" This is a descriptive cross-sectional study that assesses the influence of limited mobility on physical activity. In addition, other factors such as sleep patterns, diet, consumption of something are also assessed. All these factors were related to the age and sex of the subjects.
There are many minor and major issues I'd like the authors resolve.
Abstract
1-Add the study design to the abstract.
2- Change the keywords. Delete the words "COVID-19 pandemic lockdown" "types of physical activity" "sleep patterns" "; food and alcohol consumption" "age" and "gender". Not found in the MeSH (Medical Subject Headings). Change to "COVID-19 " "Pandemics" "Exercisa"......
Introduction
3-It is recommended that this section be expanded. It is recommended to include the objective of the study at the end of the introduction.
Materials and Methods
4-Study size: Explain how the study size was arrived at. There is a big difference in the sample between before and after.
5- It is recommended to include a section indicating the design of the study as well as the code of ethics.
6- The two periods of recruitment reflect very different situations.
October 2019 to June 2020: From normality to the first wave of the pandemic
November 2020 to June 2021 March: Includes the second and third waves of the pandemic.
However, the abstract focuses mainly on the first wave. I think this may create confusion. I recommend that the authors explain this situation.
Results
- adequate
Discussion
7-The limitations of the study design should be added as limitations. It is a descriptive cross-sectional study. Caution should be exercised with regard to cause-effect.
Conclusion
- adequate
Reference:
- adequate
Author Response
November 2, 2021
Dear Reviewer,
Thank you for the opportunity to revise and resubmit the manuscript, “How COVID-19 Pandemic Change People’s Physical Activity Distribution, Eating, and Alcohol Consumption Habits as Well as Body Mass Index?” for consideration in the International Journal of Environmental Research and Public Health.
We have submitted a revised version of the manuscript that addresses reviewer's comments. In this letter, we address each of the points raised and point to the specific change, or in rare cases, our reason for not making the change. The manuscript resubmitted has changes highlighted, as requested. We appreciate the effort reviewers that we believe will strengthen this manuscript.
Thank you.

Reviewer 2 Report
Although the authors have made considerable efforts to develop this paper, however, I believe that the current version of manuscript should be improved through some minor and re-writing. I want to provide some suggestions for the improvement of this paper as follows.
Introduction
- I think that the overall structure and writing of introduction part are not clear and well-aligned because it is not easy to catch what the research questions and strategies in this paper. Please clearly describe those things. As you already knew, the introduction section is one of the most important parts not only to draw attention of readers but also to provide guidelines for them to facilitate a clear understanding of the paper.
Theories and hypotheses
Although this paper dealt with interesting phenomena, this paper did not provide the part about the novelty of study So, it is very difficult for me to be sure that the research has an enough level of theoretical value and contribution. I think that this is the critical flaw of this paper. Please provide the part in an elaborated way. Please underline the novelty of the study related to previous studies.
Strengths and Limitations of the Study
Although the authors have attempted to explain the contributions and implications of the paper, I think that the overall quality of the explanations is low. Please provide more elaborated explanations to demonstrate its theoretical and practical contributions.
Conclusions
I recommend adding more specific conclusions according with your main findings of study.
Author Response
November 2, 2021
Dear Reviewer,
Thank you for the opportunity to revise and resubmit the manuscript, “How COVID-19 Pandemic Change People’s Physical Activity Distribution, Eating, and Alcohol Consumption Habits as Well as Body Mass Index?” for consideration in the International Journal of Environmental Research and Public Health.
We have submitted a revised version of the manuscript that addresses reviewers comments. In this letter, we address each of the points raised and point to the specific change, or in rare cases, our reason for not making the change. The manuscript resubmitted has changes highlighted, as requested. We appreciate the effort reviewers that we believe will strengthen this manuscript.
Thank you.

Round 2
Reviewer 1 Report
Dears authors;
I am satified about revised version, it is suitable to be accepted now.
Kind regards.
Reviewer 2 Report
The authors improved the manuscript according ith the recommendations.